# Effect of Abiotic Conditions on Growth, Mycotoxin Production, and Gene Expression by *Fusarium fujikuroi* Species Complex Strains from Maize

**DOI:** 10.3390/toxins15040260

**Published:** 2023-04-01

**Authors:** Ting Dong, Shouning Qiao, Jianhong Xu, Jianrong Shi, Jianbo Qiu, Guizhen Ma

**Affiliations:** 1School of Environmental and Chemical Engineering, Jiangsu Ocean University, Lianyungang 222005, China; 2Jiangsu Key Laboratory for Food Quality and Safety-State Key Laboratory Cultivation Base, Ministry of Science and Technology, Nanjing 210014, China; 3Key Laboratory for Control Technology and Standard for Agro-Product Safety and Quality, Ministry of Agriculture and Rural Affairs, Nanjing 210014, China; 4Collaborative Innovation Center for Modern Grain Circulation and Safety, Nanjing 210014, China; 5Institute of Food Safety and Nutrition, Jiangsu Academy of Agricultural Sciences, Nanjing 210014, China

**Keywords:** *Fusarium fujikuroi* species complex, water activity, temperature, mycotoxin, gene expression

## Abstract

*Fusarium fujikuroi* species complex (FFSC) strains are a major concern for food quantity and quality due to their strong ability to synthesize mycotoxins. The effects of interacting conditions of water activity, temperature, and incubation time on the growth rate, toxin production, and expression level of biosynthetic genes were examined. High temperature and water availability increased fungal growth. Higher water activity was in favor of toxin accumulation. The maximum amounts of fusaric acid (FA) and fumonisin B1 (FB1) were usually observed at 20–25 °C. *F. andiyazi* could produce a higher content of moniliformin (MON) in the cool environment than *F. fujikuroi*. The expression profile of biosynthetic genes under environmental conditions varied wildly; it was suggested that these genes might be expressed in a strain-dependent manner. FB1 concentration was positively related to the expression of *FUM1*, while a similar correlation of *FUB8* and *FUB12* with FA production could be observed in *F. andiyazi*, *F. fujikuroi*, and *F. subglutinans*. This study provides useful information in the monitoring and prevention of such toxins entering the maize production chain.

## 1. Introduction

Maize is one of the most important cereal crops in the world; in addition to supplying necessary nutrients for humans and animals, it is also an important industrial raw material for the synthesis of oil, starch, bioenergy, and even medicine [1]. According to the National Bureau of Statistics in China, maize has been the largest cereal food crop in China since 2013 [2]. The national maize planting area has reached 43 million hectares, and the yield was approximately 280 million tons at the end of 2022, which accounted for more than 40% of the cereal yield in China [3].

In the process of cultivation, maize is always susceptible to fungal diseases, among which ear rot is the most important in all maize-growing regions worldwide due to its effect on production. Moreover, diseased maize grains are often contaminated with various mycotoxins produced by pathogens, resulting in a reduction in quality, which precludes them from being used as food or feed [4]. The amount and type of fungi in the kernels depend on the climatic conditions, presence of insects, and variety. The *Fusarium fujikuroi* species complex (FFSC) has been proven to be associated with maize in recent studies [5,6,7,8]. This complex includes more than 60 phylogenetically distinct species that comprise three biogeographically structured clades according to phylogenetic studies [9]. *F. proliferatum* and *F. verticillioides* have traditionally been recognized as ear rot pathogens, and recent combined studies on the etiology of the disease and phylogenetic analyses of the fungi have revealed that *F. andiyazi* [10], *F. fujikuroi* [11], *F. temperatum* [12], and *F. subglutinans* [13] are also very pathogenic to maize kernels and can cause typical symptoms.

FFSC members can produce a series of secondary metabolites in the process of infecting the host, such as fusaric acid (FA), moniliformin (MON), beauvericin (BEA), fusarin C (FUS C), fusaproliferin (FUS), and fumonisins (FBs) [14]. FBs are the notorious contaminants in maize due to their high occurrence and potential carcinogenicity [15], and they have been the focus of many research studies. Many countries have set up strict limits for FBs due to the health risk associated with the consumption of contaminated cereals. MON, a sodium or potassium salt of 1-hydroxy-cyclobut-1-ene-3,4-dione, is a highly toxic metabolite with acute toxicity to plants and extreme toxicity to various animal species. It inhibits oxidation of the tricarboxylic acid intermediate α-ketoglutarate and pyruvate dehydrogenase by interrupting pyruvate incorporation into the tricarboxylic acid cycle [16]. MON is known to be a natural contaminant in cereals worldwide [17]. In addition, the cooccurrence of mycotoxins in food and feed represents a natural trend, and the effects of mixtures of mycotoxins in feed on farm animals are yet to be fully understood. Several studies on the combined effects of MON and FB1 concluded additive or less than additive toxicity, focusing on the relative weight of specific organs, mortality, or kidney lesions [18,19]. Fusaric acid is also widely distributed in nature, as it can be produced by many *Fusarium* species. It is well recognized that FA can distort the mitochondrial membrane and inhibit adenosine triphosphate synthesis [20], yet it seems to be mildly toxic to some animals [21]. The main concern is the synergistic interactions of FA with other cooccurring mycotoxins, such as FB1 [22]. In order to protect public health and avoid trade barriers, more and more countries have set maximum levels for the most often regulated mycotoxins, including aflatoxins, trichothecenes, ochratoxin A, zearalenone, and fumonisins [23]. However, there is no related legislation on fumonisins in China at present, due to the complicated distribution of FBs [24,25].

Mycotoxin biosynthesis is a complex process with various external environmental factors forming a regulatory network, including pathway-specific and global regulators that are often responsive to carbon and nitrogen sources, pH, ambient light, and oxidative stress [26]. Water stress and temperature are the most relevant environmental factors that influence fungal growth and mycotoxin production; therefore, they are essential to understand the overall process and to predict and prevent plant diseases and mycotoxin production. The type of mycotoxin is genetically determined, highlighting the presence or absence of necessary biosynthetic genes. With the development of genomic and transcriptomic approaches, an increasing number of secondary metabolite biosynthesis pathways have been deciphered. Mycotoxin biosynthetic genes are usually clustered together. Fumonisins and fusaric acid are the products of the *FUM* (consisting of at least 17 genes) and *FUB* (consisting of at least 12 genes) clusters, respectively [27,28]. Gene expression is also influenced by abiotic factors, and *FUM* gene expression has been used to evaluate these effects. However, contradictory relationships between *FUM* transcript levels and FBs concentrations have been reported [29,30,31]. Medina et al. [32] suggested that only some elements in the pathway could be directly correlated with toxin synthesis. Information about the dynamic expression of the *FUB* cluster under various conditions is rare, and it is meaningful to determine the consistency of *FUM* and *FUB* expression profiles under the same stress.

In a situation of changing climatic conditions, information about the behavior of different fungal species related to growth patterns and toxin production in environmental stress conditions might become critical to improve the prediction and control of mycotoxin risk. Previous results were usually based on one or two species in the complex; systematic research on the common species could provide an explanation for the population characteristics of maize pathogens. The objective of this study was to compare the impact of strain, temperature, water activity, and incubation time on the fungal growth, mycotoxin production, and gene expression of FFSC strains isolated from maize samples in China.

## 2. Results

### 2.1. Effect of Environmental Conditions on Growth

The effect of water activity and temperature on the growth rate and biomass production of FFSC strains on potato dextrose agar (PDA) is shown in Figure 1. All strains exhibited similar behavior under all studied factors. Both water activity and temperature parameters significantly affected the growth rate and dry weight of mycelium of all species. The optimal conditions for growth rate and biomass accumulation for all strains were 0.99 a_w_ and 30 °C, which were the highest conditions evaluated. Both variables of all strains significantly decreased with the water activity and temperature, and all strains were able to grow at the lowest a_w_ or temperature. A multifactorial analysis of variance (ANOVA) showed significant effects on the growth and biomass accumulation for water and temperature, as well as their interaction, in all species. (Table 1 and Table 2).

### 2.2. Effect of Environmental Conditions on Mycotoxin Production

Fusaric acid was the main metabolite within FFSC evaluated strains, and it was only absent in *F. verticillioides*. All these strains generally responded to water activity similarly, and higher water stress significantly decreased the FA concentration compared to a_w_ at 0.98 or 0.99. The FA peak value of *F. andiyazi* appeared just after 1 week, which was earlier than that of the other producers. The optimal conditions were 0.98 a_w_/25 °C and 0.99 a_w_/30 °C, respectively (Figure 2). Both *F. fujikuroi* strains showed dramatic differences under variable abiotic factors. FA synthesis was induced by water stress, and the maximal concentration was reached at 20 °C/0.95 a_w_, while the other produced the largest amounts of FA at 25 °C/0.99 a_w_ (Figure 3). Temperature had a comparable effect on FA accumulation in *F. proliferatum* (Figure 4), *F. subglutinans* (Figure 5), and *F. temperatum* (Figure 6). The maximum FA levels of the three species were produced at 20 °C after 14 days of incubation when the water activity was 0.98 or 0.99. ANOVA showed statistically significant effects for all factors considered, except when considering the strain and interactions between strain and other factors (Table 3).

Fumonisin was detected in *F. fujikuroi*, *F. proliferatum*, and *F. verticillioides* strains, and higher levels of FB1 were present at 0.98/0.99 a_w_. With regard to *F. fujikuroi*, FB1 contents increased with increasing incubation time, and the highest values were reached at 21 days at 20 °C, depending on the water activity (Figure 3). Regarding *F. proliferatum*, 30 °C reduced FB1 production at each water activity. The temporal variation feature was identical to that of *F. verticillioides*, and the most advantageous conditions for FB1 synthesis were 25 °C/0.99 a_w_, with 14 and 21 days of incubation (Figure 4). At the early stage, the variable temperature had the same effect on *F. verticillioides*. When the inoculation time was extended to 14 and 21 days, FB1 increased gradually until it tended to be stable. Low temperature favored FB1 accumulation, and maximum production was obtained at 20 °C/0.99 a_w_ after 14 days of incubation for the two *F. verticillioides* strains (Figure 7). Individual environmental factors, i.e., water activity, temperature, and incubation time, and their interactions had a significant effect on FB1 levels in fumonisin producers according to ANOVA, and both *F. verticillioides* strains showed similar patterns of responses to the treatments (Table 4).

*F. andiyazi* and *F. fujikuroi* strains produced MON. *F. andiyazi* strains produced larger amounts of MON in cooler conditions compared with *F. fujikuroi*, and maximum production always occurred after 21 incubation days at 0.98 a_w_/20 °C, although there were great variations in the yield between the two evaluated strains (Figure 2). The concentrations were extremely low in the first week and continued to increase rapidly at higher water activity levels. The highest MON contents were observed at 0.99 a_w_ after 21 days of incubation by the two *F. fujikuroi* strains at 25 °C or 30 °C, respectively (Figure 3). ANOVA showed that the MON concentration was significantly influenced by all individual factors and their interactions (Table 5).

In general, *F. andiyazi* and *F. subglutinans* strains were strong producers of FA, whereas *F. verticillioides* and *F. proliferatum* strains exhibited more enhanced FB1 synthetic ability than *F. fujikuroi*. The effect of water activity and temperature on FA and FB1 production was similar as there is a significant positive relationship between the two toxin levels in *F. fujikuroi* and *F. proliferatum* (Figure 8).

### 2.3. Effect of Environmental Conditions on Gene Expression

Figure 9 shows the expression of *FUM1* and *FUM21* in fumonisin producers on PDA for 7 days in response to temperature and water activity, reflecting the difference between species. Compared with the values resolved at 25 °C and 1 a_w_, the expression of both genes of *F. fujikuroi* seemed to be slightly repressed under most conditions, and *FUM1* expression reached the highest value was at 25 °C. For *F. proliferatum*, two targets displayed distinct rules. *FUM21* was intensively induced at the lowest water activity, which was in agreement with FB1 accumulation, while the decrease in a_w_ resulted in a gradual reduction in the expression of *FUM1*, especially at 20 and 25 °C. *FUM1* and *FUM21* were more highly expressed in *F. verticillioides* strain C19216, and they were induced at 25 °C. However, there was no obvious variation in all environment factors for the other strain (C20334). The nonparametric test showed that all factors had a significant influence on the expression of *FUM21* in *F. proliferatum* and *FUM21* in *F. verticillioides* (Table 6).

*FUB8* and *FUB12* were selected to evaluate the effect of abiotic factors on FFSC strains, and the results are presented in Figure 10. *F. fujikuroi*, *F. proliferatum*, and *F. temperatum* strains had similar behavior regarding FUB expression under different conditions, indicating that *FUB8* and *FUB12* transcript levels were mainly influenced by strain and temperature, respectively. The suppression of *FUB12* expression was obvious at lower temperatures by the two *F. fujikuroi* strains, while *FUB8* expression showed the opposite trend. The expression of *FUB8* in *F. proliferatum* strain C19215 improved upon decreasing the a_w_ from 0.99 to 0.95, but the effect of the conditions in the other strain was irregular. Both target genes were more highly expressed in *F. temperatum* strain C20278, and two strains had the highest expression levels of *FUB8* and *FUB12* at 20 °C/0.99 a_w_ and 25 °C/0.98 a_w_, respectively. Temperature was the main influencing factor for *F. andiyazi* and *F. subglutinans*. *FUB8* and *FUB12* was upregulated at 30 °C in C19239, while the highest expression of the target genes in the other *F. andiyazi* strain occurred at 25 °C. Low temperature contributed to the enhanced *FUB12* mRNA synthesis in both *F. subglutinans* strains. Despite the trace amounts of FA in *F. verticillioides*, the highest temperature and water activity strongly induced *FUB8* expression in strain C19216, and the response of C20334 to the factors was different, but no rules were followed. The nonparametric test indicated that individual factors had no remarkable effect on *FUB8* and *FUB12* expression in *F. andiyazi* and *F. verticillioides*, whereas strain significantly influenced both genes in the other species (Table 7).

The significant correlation between expression levels of both *FUM* genes appeared only in the *F. verticillioides* strain C19216, while *FUB8* expression was significantly related to *FUB12* expression in *F. proliferatum* strain C20281, *F. fujikuroi* strain C19232, and both *F. andiyazi* strains. In addition, a significant positive correlation was observed between *FUM1* expression and FB1 amounts in four out of six FB1-producing strains, but *FUM21* expression was negatively correlated with this toxin. The FA concentration in *F. andiyazi* exhibited an extremely significant and positive correlation with two *FUB* genes. A similar relationship of *FUB8* and *FUB12* with FA content occurred in *F. subglutinans* and *F. fujikuroi* (Figure 8).

In summary, increased expression of mycotoxin synthetic genes could be found in most treatments with various factors that indicated the induction of water stress compared with the control (a_w_ = 1). Considering the influence of temperature and species, expression of biosynthetic genes might be strain-dependent.

## 3. Discussion

In this study, the impact of water activity and temperature in different culture periods on growth and mycotoxin production by various FFSC strains was systematically investigated. The detailed results generated for all strains exhibited similar patterns. All species could adapt to a wide range of conditions, particularly at higher temperatures and water potentials, and mycelial growth and biomass accumulation decreased as the water availability of the media and incubation temperature were reduced. These phenomena support previous studies [31,32,33,34,35,36,37]. At present, research on the response of *F. fujikuroi* to variable conditions is lacking. It can be speculated that more species in this complex should have similar characteristics according to the published literature. Moreover, *F. verticillioides* did not show significantly better performance in favorable or unfavorable environments compared with other species. This may indicate that the dominance of *F. verticillioides* in the pathogens causing maize ear rot does not stem directly from its superior ecological adaptability. The ecosystem, even in an entire plant, is a highly complex entity comprising the crop and various biotic and abiotic factors. The dominant species niche should be the result of competition with other microorganisms in the plant-associated microbiota and long-term adaptation to the host and environmental conditions.

There has been extensive work on the combined effect of water activity and temperature on mycotoxin production, and the research described here provides the first detailed study on toxic secondary metabolite accumulation on PDA medium. Fumonisins are the most important contaminants in maize due to their strong carcinogenicity and high incidence, and we found that the optimal conditions for FB1 are lower temperatures (20–25 °C) and higher water activity (0.98–0.99 a_w_). This pattern is consistent with previous studies of mycotoxin production profiles of *F. proliferatum* [38] and *F. verticillioides* [32]. Two emerging toxins, fusaric acid and moniliformin, are often ignored in current food safety monitoring due to the lack of limit standards, as well as their low occurrence and ambiguous toxic effects. It is still meaningful and valuable to examine toxin characteristics under distinct environmental factors. Although their toxicity to humans can be mild or unknown, they could enhance the effects of the other mycotoxins and induce combined toxicity once the cereal grains are contaminated with multiple mycotoxins [22,39]; furthermore, they are usually synthesized by nondominant maize pathogens, such as *F. andiyazi* and *F. temperature*, but more species could be pandemic as the field population is always in dynamic status. The dynamic change in FA accumulation was similar to that of FB1, while MON production was favored by wet and hot conditions. Our results are in agreement with the few papers published by Pena et al. [37] and Fumero et al. [36,40], who observed the maximum FA/MON production at the highest water activity and temperature.

The results generated from different studies are sometimes nonuniform or even contradictory because of diversities in the culture conditions, such as shaking or stationary cultivation, nutrients from natural cereal grains or synthetic medium, and the origin of the strain evaluated, e.g., the soil or crop, a hot or cold region. Otherwise, with the same strain, the production of different types of fumonisin could be favored by different temperatures, despite sharing the same biosynthetic pathway [32]. We cannot give a definitive conclusion about the optimal conditions for growth and/or mycotoxin production, but it is suggested that the effect of environmental factors on the phenotypic parameters could be similar or at least overlap. It is possible to restrict the pathogen itself and its metabolites at the same time.

Fumonisins are synthesized by a polyketide pathway and involve at least 16 clustered genes that encode biosynthetic enzymes and regulatory and transport proteins in *Fusarium* [28]. In the current work, we focused on the expression profile of *FUM* genes in response to temperature and moisture stress. The expression pattern of the fumonisins-producing strains studied showed strain-dependent variation. The transcriptional regulation was reported by Jurado et al. [33] and Marín et al. [34], who revealed the significant inductive effect of water and temperature stress on FB-related gene transcripts in *F. verticillioides* and *F. proliferatum*, respectively. However, unlike the previous results, the great expression of *FUM1* was usually detected at 25 °C, and the *FUM21* transcript was induced at the lowest a_w_. Contradictory results were also found in similar studies [30,35]. The differences may have stemmed from the target evaluated, because only some key genes have a direct connection with toxin biosynthesis. The location of *FUM1*, the key enzyme of fumonisin biosynthesis, can be sensed, and it is more susceptible to changes in water activity, while the other elements in the cluster remain stable. The functional diversity of *FUM* genes may lead to various intrinsic expression patterns and responses to the changing environment. Medina et al. [32] developed a predictive model of nine *FUM* genes in relation to environmental factors with a microarray analysis and showed that the relative expression of *FUM1*, *FUM11*, *FUM13*, *FUM14*, and *FUM19* is influenced by external environments. Lazzaro et al. [41] also hypothesized that *FUM21*, encoding a transcription factor, primarily plays its activation role before the other *FUM* cluster genes, while *FUM2* remains highly active for longer. In *Aspergillus flavus*, the transcript levels of the regulatory genes did not change significantly under varying conditions, and the structural genes appeared to be highly expressed only in correspondence with the greatest AFB1 production [42,43]. Therefore, it is important to determine the appropriate test time and genes to examine, allowing some preferable results to be obtained. As a reaction to abiotic stress, the activation of *Fusarium* mycotoxin biosynthetic genes has been described in several studies and is considered part of a more general phenomenon [44,45,46]. Gene clusters of some secondary metabolites, such as fujikurins, beauvericin, and trichosetin, have been proven to be silent under all conditions and can be activated only after the application of molecular techniques [47,48,49]. In the present work, there was no significant variation of *FUM* or *FUB* expression levels in *F. fujikuroi*, which could indicate that the ecological conditions used here are less important, or that the other metabolites contribute more to adaption to stress.

Fusaric acid is the product of the *FUB* cluster containing 12 necessary genes that have been deciphered in *Fusarium* [27,50]. To our knowledge, this is the first attempt to unveil the relationship between FA accumulation and *FUB* gene expression in FFSC strains under various abiotic factors. The similar effects of temperature and water activity on *FUB* and *FUM* gene expression indicate the shared regulatory pathways controlling the production of these two metabolites. It has been stated that the expression of secondary metabolite synthetic clusters in response to multiple environmental signals is modulated by a complicated network of global regulators, including components of the VELVET complex, GATA transcription factor *Csm1* [51,52]. For example, under inducing or repressing conditions, transcript levels of gibberellin, fusarins, fumonisin, fusarubin, and even the silent genes significantly increased in the lae1-overexpressing mutant [53]. Although the expression of the *FUM* and *FUB* clusters show some correlations, strain-specific differences in the production of the corresponding toxins are apparent, possibly due to the different genetic backgrounds and other regulatory elements. In *F. fujikuroi*, the transcription factor *AreB* is necessary for the expression of most mycotoxin synthetic clusters, including *FUB* and *FUM*, in high or low environments, while another transcription factor *AreA* was reported to be effective only in fumonisin and gibberellin synthesis [54,55]. Moreover, water stress achieved by the addition of glycerol to the medium is a form of osmotic pressure, and fungi have evolved a high-osmolarity glycerol mitogen-activated protein kinase pathway (HOG) to recognize various environmental stresses. The regulatory effect of HOG kinase on gene expression and toxin production needs to be taken into account.

It is also notable that there was some inconsistency between the effect of abiotic conditions on gene expression and mycotoxin production, which means that these results did not show significant correlation. Although López-Errasquín et al. [29] and Lazzaro et al. [41] identified a correlation between FB concentration and *FUM* gene transcripts, the correlation was usually found under a certain factor or specific parameters of the factor. Gene expression could mirror toxin production profiles under all conditions except temperature [35], which implied the different effects of varied conditions. In addition, Medina et al. [31] showed that the synthesis of fumonisins increased at higher transcript levels of *FUM19*, while higher expression of *FUM14* was correlated with decreased concentration of FBs. *FUM11* positively or negatively regulated the production of FB1 and FB2, respectively. The correlation of gene expression and production of the toxin seems to be highly variable because the expression of only a few genes is usually directly correlated with the toxin biosynthesis. The lack of correlation could be explained by a post-transcriptional regulatory mechanism. While transcriptional to translational correlations are often not very strong, it is essential to determine the relationship of gene expression to protein translation in fumonisin production to better understand the mechanism of biosynthesis.

In conclusion, fungal growth and mycotoxin accumulation are influenced by environmental factors. Mycotoxin contamination is the result of multiple factors in the field, including fungicide, pH, and light, and a better understanding of the optimal conditions is useful in the design of prevention and control strategies.

## 4. Materials and Methods

### 4.1. Fungal Strains

Two strains each of six species in the *Fusarium fujikuroi* species complex were isolated from maize samples in various regions of China between 2019 and 2020. These strains were identified by morphological characteristics, toxin profile, and phylogenetic analysis of partial elongation factor 1α. Single-spore cultures were stored as spore suspensions in 20% glycerol at −80 °C. Detailed information on the strains used in this study is listed in Table 8.

### 4.2. Inoculation, Incubation, and Growth Assessment

Subsequent studies were carried out with potato dextrose agar medium. Water activity was modified to 0.95, 0.98, and 0.99 by the addition of 245, 198, and 108 mL of glycerol per litre of culture medium. The accuracy of water activity was confirmed with an AquaLab Series 4TE (Decagon Devices, Inc., Pullman, WA, USA).

A 5 mm diameter agar disc from the margin of a fresh colony was transferred and inoculated onto the center of each plate. Inoculated plates were incubated at 20, 25, and 30 °C for 7, 14, and 21 days. Three biological replicates were conducted for each experimental condition. Two diameters of the colonies were measured at right angles every day until the colony reached the edge of the plate. The radii of the colonies were plotted against time, and linear regression was used to obtain the growth rate (mm/day). After 7 days of incubation, the whole mycelium was removed, frozen at −80 °C, lyophilized, and weighed as the biomass.

### 4.3. Mycotoxin Analysis

After 7, 14, and 21 days, agar plates were weighed, ground, and shaken in 20 mL of acetonitrile/water (4:1) at 180 rpm for 2 h. Then, the extract was filtered through Whatman No. 4 filter paper, and an aliquot (4 mL) of the supernatant was transferred to a glass tube until evaporation to dryness under N_2_ flow. The dried samples were resuspended in 1 mL of acetonitrile, filtered through a 0.22 µm Millipore membrane, and analyzed using an LC-20ADXR liquid chromatograph (Shimadzu, Kyoto, Japan) coupled to an AB SCIEX Triple-Quad mass spectrometer (Applied Biosystems, Foster City, CA, USA).

The analytical column was a Kinetex 100A C18 column (100 × 2.3 mm, 2.6 μm) from Phenomenex (Torrance, CA, USA), and the column temperature was maintained at 40 °C. The flow rate was 0.5  mL/min, and the injection volume was 2 μL. The mass spectrometric analyses were performed with the following operation parameters: gas temperature, 500 °C; gas flow rate, 10  L/min; nebulizer gas pressure, 50  psi; capillary voltage, 5500  V. Nitrogen was used as the ion source and in the collision cell. Mycotoxins were analyzed via multiple reaction monitoring (MRM).

The mobile phase consisted of 5 mM ammonium acetate/acetic acid (99.9/0.1, v/v) (A) and methanol (B). The stepwise high-performance liquid chromatography conditions were as follows: 0–0.01 min, solvent A linearly increased to 10%; 1.6–2 min, solvent A linearly increased from 10% to 35%; 2–4.44 min, solvent A linearly increased to 55%; 4.44–6.5 min, solvent A linearly increased from 55% to 90%; 6.5–12 min, solvent A remained constant at 90%; 12–15 min, solvent A remained constant at 10%. The values of the limit of quantitation and limit of detection were 10 and 5 µg/kg, respectively.

### 4.4. Gene Expression Analysis

Total RNA was extracted from 7 day old mycelia of *Fusarium* strains grown on PDA with or without glycerol using an Axyprep multisource total RNA miniprep kit (Axygen Scientific, Inc., Union City, CA, USA). The amount and quality of RNA were estimated by Nanodrop (Thermo Fisher Scientific, Waltham, MA, USA). First-strand cDNA was obtained by the Prime ScriptTM RT Master Mix Kit (Takara Co., Ltd., Dalian, China) according to the manufacturer’s instructions and stored at −80 °C until use in PCR.

Universal primers were designed with Primer Premier 5 software (PREMIER Biosoft International, Palo Alto, California, USA) on exon–exon junctions in the target mRNA to avoid the amplification of genomic DNA. Primers QFFFUB8F (5′–CTSTTCCAYGTTGCTGGWATMTG–3′)/QFFFUB8R (5′–CTTCMGAGTARGGACGCATCTC–3′) and QFFFub12F (5′–GCMTCTTTCTTYTCAAAGGC–3′)/QFFFub12R (5′–CTGTTATGYAAAACCCACCA–3′) were used to amplify *FUB8* and *FUB12* in every species studied. Primers QFFFUM1F1 (5′–TACCACTCWCATCACATGCAAG–3′)/QFFFUM1R1 (5′–TACRGGRCTCTCCAGATTTTG–3′) and QFFFUM21F1 (5′–ATTRCCGYTGATCCACGACGA–3′)/QFFFUM21R1 (5′–AGCTYGCGCTMTRCTSAGASG–3′) were applied in the amplification of *FUM1* and *FUM21* in *F. verticillioides*, *F. fujikuroi*, and *F. proliferatum*, respectively. The actin gene was selected as the internal control, and the primers QFFActF (5′–CCACCAGACATGACAATGTT–3′)/QFFActR (5′–CGTGATCTTACCGACTACCTC–3′) were used for all stains. Real-time PCR was conducted by TB Green Premix Ex Taq (Takara Co., Ltd., Dalian, China) according to the manufacturer’s protocol. Real-time PCR reactions were performed in a LightCycler 96 system (Roche, Basel, Switzerland), and the data were analyzed with sequence detector software. Gene expression profiles were normalized to actin expression, and relative changes in gene expression levels were calculated using the 2^−∆∆Ct^ method.

### 4.5. Statistical Analysis

The statistical effects of the individual factor and their interactions on growth rate, biomass accumulation, and mycotoxin production were evaluated by ANOVA using GraphPad Prism version 8.3.0 (GraphPad Software, San Diego, CA, USA). Relative expression levels of mycotoxin synthetic genes under different conditions were analyzed by nonparametric tests for multiple comparisons using SPSS Statistics 26.0 (SPSS Inc., Chicago, IL, USA).

## Figures and Tables

**Figure 1 toxins-15-00260-f001:**
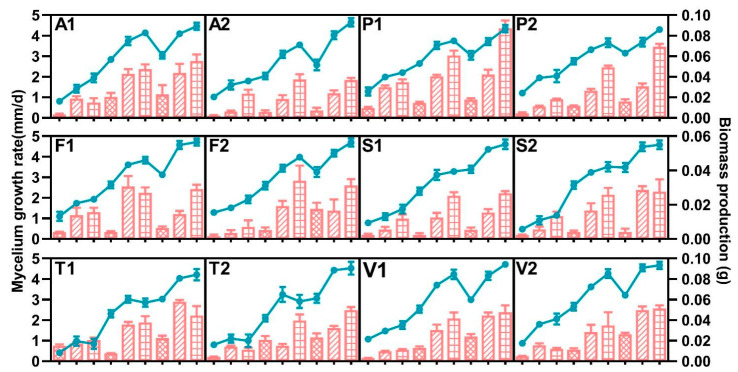
Effect of a_W_ and temperature on growth rate (line) and biomass production (column) of *Fusarium andiyazi* (A1: C19239; A2: C20180), *F. fujikuroi* (F1: C19232; F2: C20076), *F. proliferatum* (P1: C19215; P2: C20281), *F. subglutinans* (S1: C19068; S2: C20215), *F. temperatum* (T1: C20277; T2: C20278), and *F. verticillioides* (V1: C19216; V2: C20334) strains. The first, middle, and last three points and columns represent the results at 20 °C, 25 °C, and 30 °C, respectively. The columns filled with grids, oblique lines, and squares represent the results under 0.95, 0.98, and 0.99 a_w_, respectively. Error bars represent the standard error measured between independent replicates.

**Figure 2 toxins-15-00260-f002:**
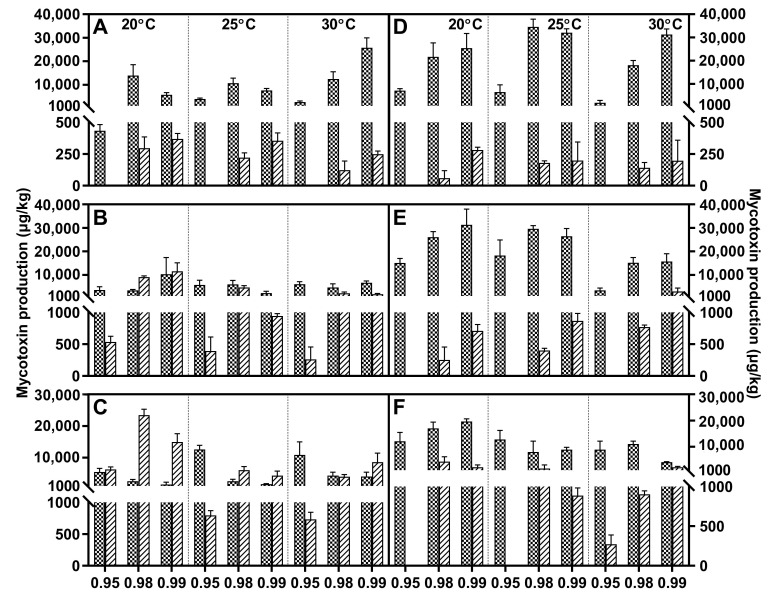
Fusaric acid (grid) and moniliformin (oblique line) levels produced by *Fusarium andiyazi* strain A1 (**A**–**C**) and A2 (**D**–**F**) growing on PDA adjusted to different water activity, temperature, and incubation time: 7 days (**A**,**D**), 14 days (**B**,**E**), and 21 days (**C**,**F**). Error bars represent the standard error measured between independent replicates.

**Figure 3 toxins-15-00260-f003:**
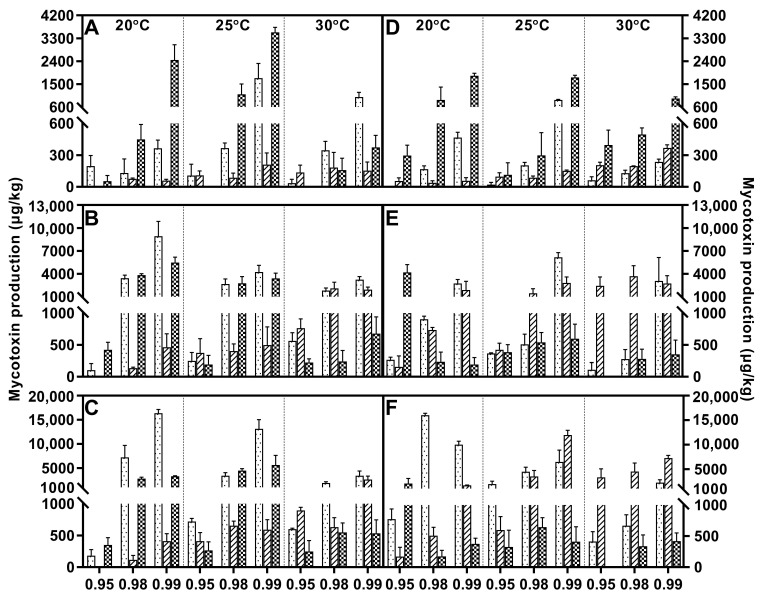
Fusaric acid (grid), fumonisin B1 (black dot) and moniliformin (oblique line) levels produced by *Fusarium fujikuroi* strain F1 (**A**–**C**) and F2 (**D**–**F**) growing on PDA adjusted to different water activity, temperature, and incubation time: 7 days (**A**,**D**), 14 days (**B**,**E**), and 21 days (**C**,**F**). Error bars represent the standard error measured between independent replicates.

**Figure 4 toxins-15-00260-f004:**
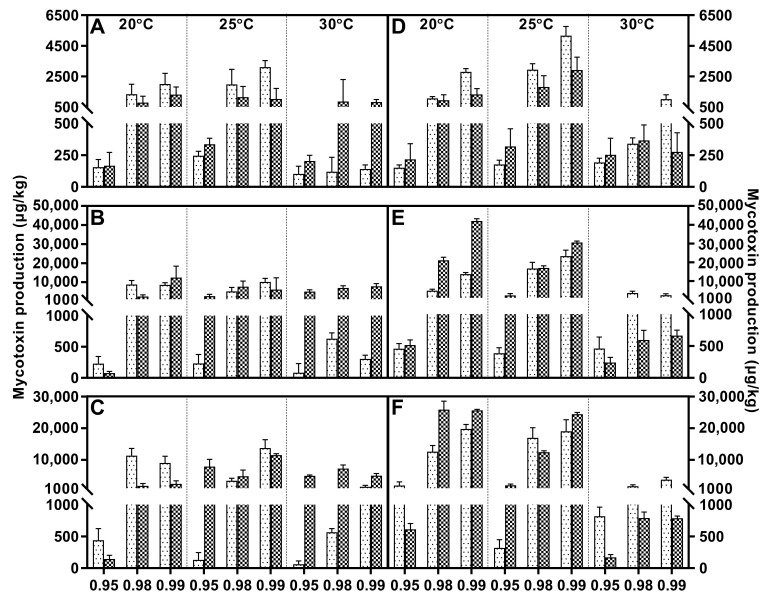
Fusaric acid (grid) and fumonisin B1 (black dot) levels produced by *Fusarium proliferatum* strain P1 (**A**–**C**) and P2 (**D**–**F**) growing on PDA adjusted to different water activity, temperature, and incubation time: 7 days (**A**,**D**), 14 days (**B**,**E**), and 21 days (**C**,**F**). Error bars represent the standard error measured between independent replicates.

**Figure 5 toxins-15-00260-f005:**
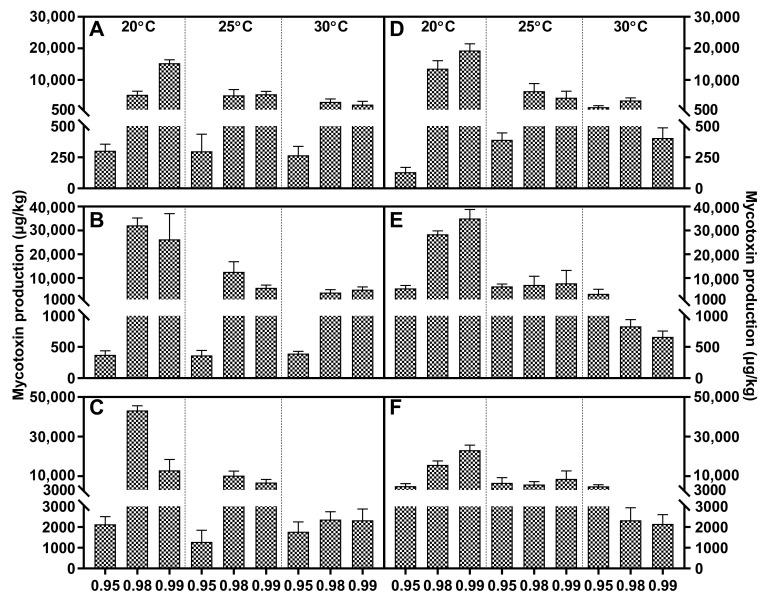
Fusaric acid levels produced by *Fusarium subglutinans* strain S1 (**A**–**C**) and S2 (**D**–**F**) growing on PDA adjusted to different water activity, temperature, and incubation time: 7 days (**A**,**D**), 14 days (**B**,**E**), and 21 days (**C**,**F**). Error bars represent the standard error measured between independent replicates.

**Figure 6 toxins-15-00260-f006:**
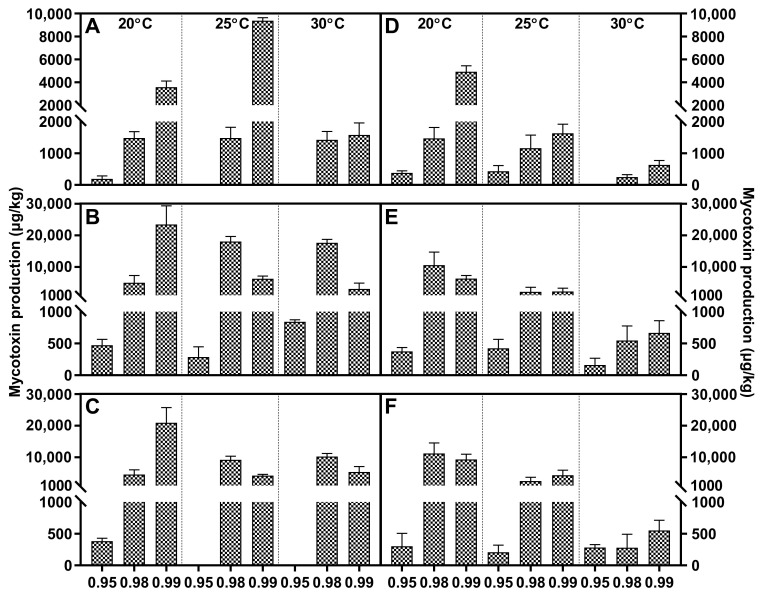
Fusaric acid levels produced by *Fusarium temperatum* strain T1 (**A**–**C**) and T2 (**D**–**F**) growing on PDA adjusted to different water activity, temperature, and incubation time: 7 days (**A**,**D**), 14 days (**B**,**E**), and 21 days (**C**,**F**). Error bars represent the standard error measured between independent replicates.

**Figure 7 toxins-15-00260-f007:**
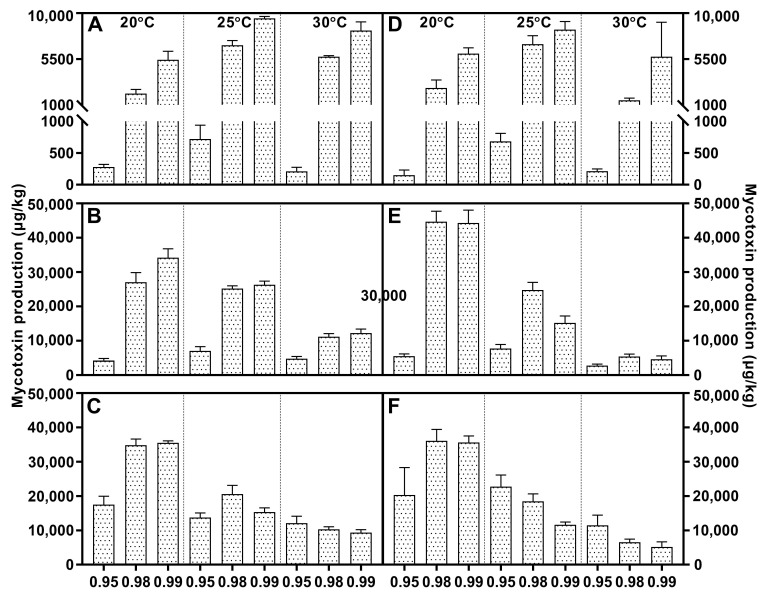
Fumonisin B1 levels produced by *Fusarium verticillioides* strain V1 (**A**–**C**) and V2 (**D**–**F**) growing on PDA adjusted to different water activity, temperature, and incubation time: 7 days (**A**,**D**), 14 days (**B**,**E**), and 21 days (**C**,**F**). Error bars represent the standard error measured between independent replicates.

**Figure 8 toxins-15-00260-f008:**
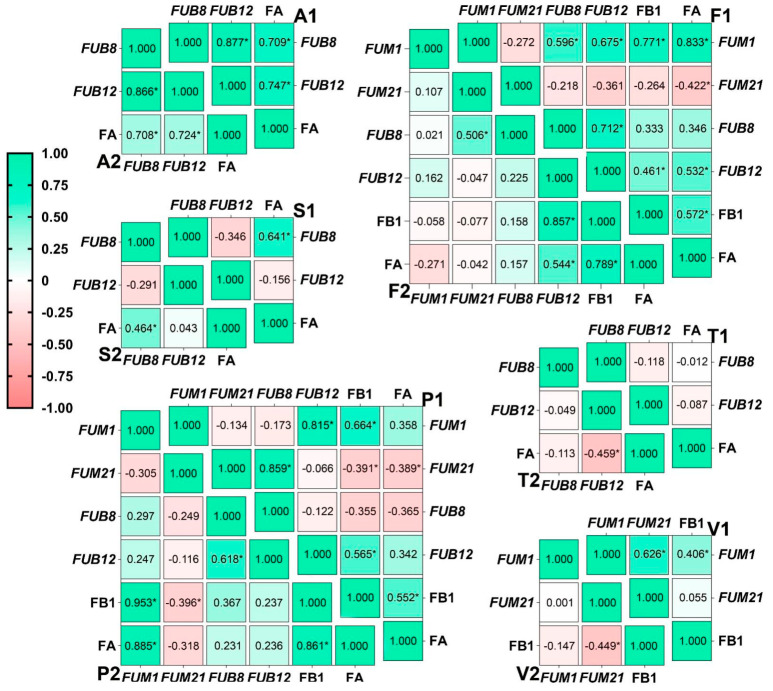
The heatmap of the pairwise Pearson’s correlation between variables of *Fusarium* strains incubated for 7 days on PDA plate. The strains included *Fusarium andiyazi* (A1: C19239; A2: C20180), *F. fujikuroi* (F1: C19232; F2: C20076), *F. proliferatum* (P1: C19215; P2: C20281), *F. subglutinans* (S1: C19068; S2: C20215), *F. temperatum* (T1: C20277; T2: C20278), and *F. verticillioides* (V1: C19216; V2: C20334). *FUM1* and *FUM21* coded the polyketide synthase and Zn(II)2Cys6DNA-binding transcription factor in fumonisin synthetic cluster, respectively. *FUB8* and *FUB12* code the non-ribosmal peptide synthetase-like carboxylic acid reductase and Zn(II)2Cys6DNA-binding transcription factor in the fusaric acid synthetic cluster, respectively. FB1 and FA denote fumonisin B1 and fusaric acid, respectively. * *p* < 0.05.

**Figure 9 toxins-15-00260-f009:**
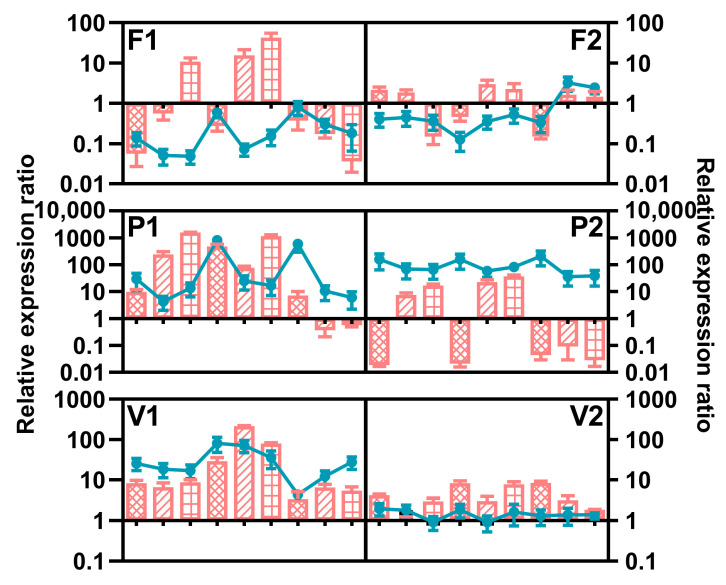
Relative expression of *FUM1* (line) and *FUM21* (column) of *Fusarium fujikuroi* (F1: C19232; F2: C20076), *F. proliferatum* (P1: C19215; P2: C20281), and *F. verticillioides* (V1: C19216; V2: C20334) strains incubated for 7 days under various conditions. The measured quantity of cDNA was normalized using the Cq values obtained for *Act* cDNA amplifications. The values represent the expression level of the target gene in the different culture conditions relative to the control culture (25 °C, a_W_ = 1). The data represent the means of three independent repetitions. Error bars represent the standard error measured between independent replicates. The first, middle, and last three points and columns represent the results at 20 °C, 25 °C, and 30 °C, respectively. The columns filled with grids, oblique lines, and squares represent the results under 0.95, 0.98, and 0.99 a_w_, respectively. Error bars represent the standard error measured between independent replicates.

**Figure 10 toxins-15-00260-f010:**
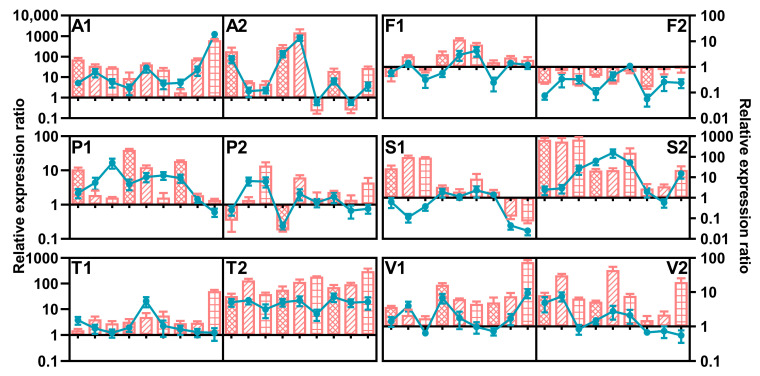
Relative expression of *FUB8* (line) and *FUB12* (column) of *Fusarium andiyazi* (A1: C19239; A2: C20180), *F. fujikuroi* (F1: C19232; F2: C20076), *F. proliferatum* (P1: C19215; P2: C20281), *F. subglutinans* (S1: C19068; S2: C20215), *F. temperatum* (T1: C20277; T2: C20278), and *F. verticillioides* (V1: C19216; V2: C20334) strains incubated for 7 days under various conditions. The measured quantity of cDNA was normalized using the Cq values obtained for *Act* cDNA amplifications. The values represent the expression level of the target gene in the different culture conditions relative to the control culture (25 °C, a_W_ = 1). The data represent the means of three independent repetitions. Error bars represent the standard error measured between independent replicates. The first, middle, and last three points and columns represent the result at 20 °C, 25 °C, and 30 °C, respectively. The columns filled with grids, oblique lines, and squares represent the results under 0.95, 0.98, and 0.99 a_w_, respectively.

**Table 1 toxins-15-00260-t001:** Analysis of variance on the effects of different strains (S), water activity (a_w_), temperature (T), and their interactions on growth rate of *Fusarium fujikuroi* species complex strains on PDA plate.

Source of Variation	*F. andiyazi*			*F. fujikuroi*			*F. proliferatum*			*F. subglutinans*			*F. temperatum*			*F. verticillioides*		
	df	F	MS	df	F	MS	df	F	MS	df	F	MS	df	F	MS	df	F	MS
S	1	84.487 *	1.327	1	1.002	0.025	1	0.579	0.009	1	0.019	0.001	1	6.564 *	0.415	1	61.880 *	1.015
a_w_	2	3584.760 *	56.315	2	2369.382 *	58.365	2	2485.337 *	37.368	2	4643.667 *	95.257	2	1333.440 *	84.275	2	4064.686 *	66.647
T	2	1144.040 *	17.972	2	606.962 *	14.951	2	695.150 *	10.452	2	407.928 *	8.368	2	129.193 *	8.165	2	1224.686 *	20.081
S × a_w_	2	92.893	1.459	2	2.228	0.055	2	1.751	0.026	2	28.363 *	0.582	2	1.793	0.113	2	14.637 *	0.240
S × T	2	8.156 *	0.128	2	11.987 *	0.295	2	10.279 *	0.155	2	0.175	0.004	2	0.482	0.030	2	10.490 *	0.172
a_w_ × T	4	39.299 *	0.617	4	26.796 *	0.660	4	12.725 *	0.191	4	23.990 *	0.492	4	13.069 *	0.826	4	41.187 *	0.675
S × a_w_ × T	4	14.184 *	0.223	4	1.170	0.029	4	1.191	0.018	4	1.454	0.030	4	1.514	0.096	4	5.030 *	0.082

* Significant *p* < 0.01; df: degrees of freedom; F: Snedecor-F; MS: mean square.

**Table 2 toxins-15-00260-t002:** Analysis of variance on the effects of different strains (S), water activity (a_w_), temperature (T), and their interactions on biomass accumulation of *Fusarium fujikuroi* species complex strains on PDA plate.

Source of Variation	*F. andiyazi*			*F. fujikuroi*			*F. proliferatum*			*F. subglutinans*			*F. temperatum*			*F. verticillioides*		
	df	F	MS	df	F	MS	df	F	MS	df	F	MS	df	F	MS	df	F	MS
S	1	177.033 *	0.003	1	0.392	0.001	1	72.807 *	0.003	1	6.794 *	0.001	1	14.267 *	0.001	1	0.526	0.001
a_w_	2	211.398 *	0.004	2	30.937 *	0.001	2	139.288 *	0.005	2	52.366 *	0.001	2	75.640 *	0.004	2	129.654 *	0.008
T	2	271.129 *	0.005	2	16.586 *	0.001	2	333.793 *	0.013	2	323.153 *	0.002	2	44.037 *	0.002	2	50.965 *	0.003
S × a_w_	2	35.106 *	0.001	2	4.338 *	0.001	2	0.883	0.001	2	3.964 *	0.001	2	2.208	0.001	2	1.827	0.001
S × T	2	16.171 *	0.001	2	2.794	0.001	2	10.701 *	0.000	2	10.620 *	0.001	2	11.134 *	0.001	2	0.440	0.001
a_w_ × T	4	9.384 *	0.001	4	3.435 *	0.001	4	35.363 *	0.001	4	26.373 *	0.001	4	8.796 *	0.001	4	5.466 *	0.001
S × a_w_ × T	4	3.596 *	0.001	4	3.460	0.001	4	0.676	0.001	4	8.383 *	0.001	4	8.960 *	0.001	4	0.087	0.001

* Significant *p* < 0.01; df: degrees of freedom; F: Snedecor-F; MS: mean square.

**Table 3 toxins-15-00260-t003:** Analysis of variance on the effects of strain (S), water activity (a_w_), temperature (T), incubation time (I), and their interactions on fusaric acid production by *Fusarium fujikuroi* species complex strains on PDA plate.

Source of Variation	*F. andiyazi*			*F. fujikuroi*			*F. proliferatum*			*F. subglutinans*			*F. temperatum*		
	df	F	MS	df	F	MS	df	F	MS	df	F	MS	df	F	MS
S	1	283.253 *	4.498 × 10^9^	1	64.840 *	3.497 × 10^7^	1	244.883 *	6.994 × 10^8^	1	0.491	7.803 × 10^6^	1	61.901 *	4.077 × 10^8^
a_w_	2	46.065 *	7.316 × 10^8^	2	40.200 *	2.168 × 10^7^	2	325.476 *	9.296 × 10^8^	2	80.839 *	1.286 × 10^9^	2	83.057 *	5.471 × 10^8^
T	2	5.482 *	8.706 × 10^7^	2	49.426 *	2.665 × 10^7^	2	176.286 *	5.035 × 10^8^	2	167.441 *	2.664 × 10^9^	2	24.695 *	1.627 × 10^8^
I	2	36.490 *	5.795 × 10^8^	2	7.334 *	3.955 × 10^6^	2	376.419 *	1.075 × 10^9^	2	26.050 *	4.144 × 10^8^	2	32.805 *	2.161 × 10^8^
S × a_w_	2	28.696 *	4.557 × 10^8^	2	50.772 *	2.738 × 10^7^	2	135.728 *	3.877 × 10^8^	2	11.628 *	1.850 × 10^8^	2	16.350 *	1.077 × 10^8^
S × T	2	31.014 *	4.925 × 10^8^	2	21.302 *	1.149 × 10^7^	2	258.669 *	7.388 × 10^8^	2	0.274	4.356 × 10^6^	2	3.477 *	2.290 × 10^7^
S × I	2	15.001 *	2.382 × 10^8^	2	12.886 *	6.949 × 10^6^	2	59.652 *	1.704 × 10^8^	2	1.333	2.120 × 10^7^	2	12.038 *	7.930 × 10^7^
a_w_ × T	4	3.664 *	5.818 × 10^7^	4	9.254 *	4.990 × 10^6^	4	73.850 *	2.109 × 10^8^	4	41.383 *	6.583 × 10^8^	4	20.105 *	1.324 × 10^8^
a_w_ × I	4	37.044 *	5.883 × 10^8^	4	2.873 *	1.549 × 10^6^	4	76.443 *	2.183 × 10^8^	4	4.342 *	6.908 × 10^7^	4	10.866 *	7.158 × 10^7^
T × I	4	7.559 *	1.200 × 10^8^	4	7.911 *	4.266 × 10^6^	4	41.091 *	1.174 × 10^8^	4	10.955 *	1.743 × 10^8^	4	4.342 *	2.860 × 10^7^
S × a_w_ × T	4	1.654	2.627 × 10^7^	4	14.865 *	8.016 × 10^6^	4	44.084 *	1.259 × 10^8^	4	5.210 *	8.289 × 10^7^	4	20.633 *	1.359 × 10^8^
S × a_w_ × I	4	1.173	1.863 × 10^7^	4	7.362 *	3.970 × 10^6^	4	35.331 *	1.009 × 10^8^	4	7.537 *	1.199 × 10^8^	4	3.127 *	2.060 × 10^7^
S × T × I	4	4.912 *	7.801 × 10^7^	4	2.588 *	1.396 × 10^6^	4	65.870 *	1.881 × 10^8^	4	3.064 *	4.874 × 10^7^	4	0.937	6.173 × 10^6^
a_w_ × T × I	8	5.666 *	8.999 × 10^7^	8	2.671 *	6.831 × 10^6^	8	29.330 *	8.377 × 10^7^	8	5.800 *	9.226 × 10^7^	8	4.584 *	3.019 × 10^7^
S × a_w_ × T × I	8	1.364	2.165 × 10^7^	8	5.004 *	2.699 × 10^6^	8	11.696 *	3.340 × 10^7^	8	3.498 *	5.565 × 10^7^	8	9.041 *	5.956 × 10^7^

* Significant *p* < 0.01; df: degrees of freedom; F: Snedecor-F; MS: mean square.

**Table 4 toxins-15-00260-t004:** Analysis of variance on the effects of strain (S), water activity (a_w_), temperature (T), incubation time (I), and their interactions on fumonisin production by *Fusarium fujikuroi*, *F. proliferatum*, and *F. verticillioides* strains on PDA plate.

Source of Variation	*F. fujikuroi*			*F. proliferatum*			*F. verticillioides*		
	df	F	MS	df	F	MS	df	F	MS
S	1	8.009 *	1.609 × 10^7^	1	114.647 *	2.952 × 10^8^	1	0.232	1.700 × 10^6^
a_w_	2	126.308 *	2.537 × 10^8^	2	294.806 *	7.592 × 10^8^	2	193.393 *	1.417 × 10^7^
T	2	48.340 *	9.709 × 10^7^	2	197.072 *	5.075 × 10^8^	2	325.243 *	2.383 × 10^7^
I	2	146.829 *	2.949 × 10^8^	2	161.533 *	4.160 × 10^8^	2	481.660 *	3.529 × 10^7^
S × a_w_	2	13.125 *	2.636 × 10^7^	2	25.346 *	6.527 × 10^7^	2	5.666 *	4.151 × 10^7^
S × T	2	0.001	7.669 × 10^2^	2	24.966 *	6.429 × 10^7^	2	24.826	1.819 × 10^8^
S × I	2	1.754	3.523 × 10^6^	2	19.119 *	4.923 × 10^7^	2	0.519	3.800 × 10^6^
a_w_ × T	4	15.294 *	3.072 × 10^7^	4	56.099 *	1.445 × 10^8^	4	57.532 *	4.215 × 10^8^
a_w_ × I	4	31.036 *	6.234 × 10^7^	4	37.923 *	9.766 × 10^7^	4	51.646 *	3.784 × 10^8^
T × I	4	30.765 *	6.179 × 10^7^	4	30.665 *	7.896 × 10^7^	4	96.912 *	7.100 × 10^8^
S × a_w_ × T	4	6.606 *	1.327 × 10^7^	4	15.778 *	4.063 × 10^7^	4	6.327 *	4.635 × 10^7^
S × a_w_ × I	4	11.726 *	2.355 × 10^7^	4	4.182 *	1.077 × 10^7^	4	4.257 *	3.118 × 10^7^
S × T × I	4	4.315 *	8.666 × 10^6^	4	8.442 *	2.174 × 10^7^	4	10.196 *	7.470 × 10^7^
a_w_ × T × I	8	9.219 *	1.852 × 10^7^	8	8.389 *	2.160 × 10^7^	8	19.139 *	1.402 × 10^8^
S × a_w_ × T × I	8	5.032 *	1.011 × 10^7^	8	5.470 *	1.408 × 10^7^	8	2.347 *	1.719 × 10^7^

* Significant *p* < 0.01; df: degrees of freedom; F: Snedecor-F; MS: mean square.

**Table 5 toxins-15-00260-t005:** Analysis of variance on the effects of strain (S), water activity (a_w_), temperature (T), incubation time (I), and their interactions on moniliformin production by *Fusarium andiyazi* and *F. fujikuroi* on PDA plate.

Source of Variation	*F. andiyazi*			*F. fujikuroi*		
	df	F	MS	df	F	MS
S	1	165.819 *	3.793 × 10^8^	1	201.331 *	7.644 × 10^7^
a_w_	2	53.713 *	1.229 × 10^8^	2	74.975 *	2.847 × 10^7^
T	2	62.539 *	1.431 × 10^8^	2	86.057 *	3.267 × 10^7^
I	2	112.895 *	2.583 × 10^8^	2	155.341 *	5.898 × 10^7^
S × a_w_	2	20.811 *	4.761 × 10^7^	2	34.034 *	1.292 × 10^7^
S × T	2	52.985 *	1.212 × 10^8^	2	23.524 *	8.932 × 10^6^
S × I	2	57.470 *	1.315 × 10^8^	2	81.936 *	3.111 × 10^7^
a_w_ × T	4	10.817 *	2.474 × 10^7^	4	9.711 *	3.687 × 10^6^
a_w_ × I	4	14.864 *	3.400 × 10^7^	4	35.043 *	1.331 × 10^7^
T × I	4	23.853 *	5.457 × 10^7^	4	29.757 *	1.130 × 10^7^
S × a_w_ × T	4	5.738 *	1.313 × 10^7^	4	16.148 *	6.131 × 10^6^
S × a_w_ × I	4	5.619 *	1.286 × 10^7^	4	20.626 *	7.832 × 10^6^
S × T × I	4	15.636 *	3.577 × 10^7^	4	16.677 *	6.332 × 10^6^
a_w_ × T × I	8	6.476 *	1.481 × 10^7^	8	8.784 *	3.335 × 10^6^
S × a_w_ × T × I	8	3.991 *	9.129 × 10^6^	8	10.463 *	3.973 × 10^6^

* Significant *p* < 0.01; df: degrees of freedom; F: Snedecor-F; MS: mean square.

**Table 6 toxins-15-00260-t006:** The result of nonparametric analysis of *FUM1* and *FUM21* expression under different conditions.

*FUM1*	N	Mean ± SD	a_w_		Temperature		Strain	
			*H*	*P*	*H*	*P*	*Z*	*P*
*F. fujikuroi*	54	2.43 ± 3.03	5.836	0.054	3.125	0.210	−2.829	0.005
*F. proliferatum*	54	247.78 ± 427.05	4.728	0.296	25.129	0.000	−2.638	0.008
*F. verticillioides*	54	16.53 ± 31.12	15.569	0.000	13.151	0.001	−3.555	0.000
** *FUM21* **	**N**	**Mean ± SD**	**a_w_**		**Temperature**		**Strain**	
			** *H* **	** *P* **	** *H* **	** *P* **	** *Z* **	** *P* **
*F. fujikuroi*	54	2.91 ± 2.96	14.875	0.001	8.255	0.016	−1.055	0.291
*F. proliferatum*	54	118.17 ± 204.60	9.078	0.002	6.735	0.034	−3.694	0.000
*F. verticillioides*	54	17.86 ± 25.57	2.056	0.358	0.842	0.656	−6.306	0.000

SD: standard deviation; H: Kruskal–Wallis test; Z: Mann–Whitney test.

**Table 7 toxins-15-00260-t007:** The result of nonparametric analysis of *FUB8* and *FUB12* expression under different conditions.

*FUB8*	N	Mean ± SD	a_w_		Temperature		Strain	
			*H*	*P*	*H*	*P*	*Z*	*P*
*F. andiyazi*	54	189.99 ± 389.46	1.37	0.504	2.737	0.254	−1.047	0.295
*F. fujikuroi*	54	2.02 ± 2.97	5.347	0.069	6.528	0.038	−5.216	0.000
*F. proliferatum*	54	6.80 ± 9.77	1.009	0.411	0.269	0.874	−2.673	0.008
*F. subglutinans*	54	353.82 ± 1109.92	1.448	0.485	35.219	0.000	−3.99	0.001
*F. temperatum*	54	65.16 ± 84.90	6.034	0.049	3.658	0.161	−5.821	0.000
*F. verticillioides*	54	9.74 ± 9.90	2.667	0.264	4.376	0.112	−1.791	0.073
** *FUB12* **	**N**	**Mean ± SD**	**a_w_**		**Temperature**		**Strain**	
			** *H* **	** *P* **	** *H* **	** *P* **	** *Z* **	** *P* **
*F. andiyazi*	54	80.76 ± 229.14	2.376	0.305	4.224	0.121	−0.268	0.789
*F. fujikuroi*	54	0.78 ± 0.86	16.753	0.000	3.723	0.155	−4.593	0.000
*F. proliferatum*	54	3.50 ± 4.25	0.374	1.000	4.987	0.083	−3.746	0.000
*F. subglutinans*	54	16.17 ± 31.82	2.67	0.263	14.821	0.001	−4.922	0.000
*F. temperatum*	54	8.87 ± 7.99	4.079	0.130	0.429	0.807	−6.116	0.000
*F. verticillioides*	54	2.55 ± 2.58	4.712	0.095	4.01	0.135	−1.237	0.216

SD: standard deviation; H: Kruskal–Wallis test; Z: Mann–Whitney test.

**Table 8 toxins-15-00260-t008:** Origin of the *Fusarium fujikuroi* species complex strains used in this study.

Strain	Code Name	Species	Host	Location	Year
C19239	A1	*Fusarium andiyazi*	Maize	Guizhou Province, southwestern China	2019
C20180	A2	*Fusarium andiyazi*	Maize	Heliongjiang Province, northeastern China	2020
C19232	F1	*Fusarium fujikuroi*	Maize	Guizhou Province, southwestern China	2019
C20076	F2	*Fusarium fujikuroi*	Maize	Hubei Province, central China	2020
C19215	P1	*Fusarium proliferatum*	Maize	Guizhou Province, southwestern China	2019
C20281	P2	*Fusarium proliferatum*	Maize	Heliongjiang Province, northeastern China	2020
C19068	S1	*Fusarium subglutinans*	Maize	Heliongjiang Province, northeastern China	2019
C20215	S2	*Fusarium subglutinans*	Maize	Shaanxi Province, northwestern China	2020
C20277	T1	*Fusarium temperatum*	Maize	Yunnan Province, southwestern China	2020
C20278	T2	*Fusarium temperatum*	Maize	Yunnan Province, southwestern China	2020
C19216	V1	*Fusarium verticillioides*	Maize	Henan Province, central China	2019
C20334	V2	*Fusarium verticillioides*	Maize	Jiangsu Province, eastern China	2020

## Data Availability

Not applicable.

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
