# Peer review of "Effect of Abiotic Conditions on Growth, Mycotoxin Production, and Gene Expression by Fusarium fujikuroi Species Complex Strains from Maize"

_toxins, 2023, doi:10.3390/toxins15040260_

Round 1
Reviewer 1 Report
Line 55
Please add TCA description since it the first time it appears (eg tricarboxylic acid)
Line 118
Please add PDA description since it the first time it appears (eg potato dextrose agar)
Line 309
… are easily ignored in current food safety monitoring.
Please explain the reasons (eg legislation etc)
Author Response
Response to Reviewer 1 Comments
Point 1: Please add TCA description since it the first time it appears (eg tricarboxylic acid)
Response 1: Thanks very much for your advice. We have revised this part as suggested.
Point 2: Line 118 Please add PDA description since it the first time it appears (eg potato dextrose agar)
Response 2: Thanks very much for your advice. We have revised this part as suggested in the revised manuscript (Line 102, Page 3).
Point 3: Line 309… are easily ignored in current food safety monitoring. Please explain the reasons (eg legislation etc)
Response 3: We have revised this part according to the suggestion.
Reviewer 2 Report
The authors have done several analyses and presented them in the form of tables and figures. A major lacuna is the figure captions are not self-explanatory. Readers will not search through the entire 25-page manuscript to find the details of abbreviations used in any figure. Thus, all the abbreviations used in any figure should be explained in the figure caption itself.
This study could have more practical utility if temperatures more than 30°C should have also been considered. See the fig 1 indicated that the pattern is not yet reached the plateau, which means, if 35, 40, or 45 °C temperatures would have chosen, it could have given more clear indication that which temperature gives the highest biomass and toxin production. It is possible that >30 °C could give even higher biomass, mycelial growth, and toxin production (most probably as the data projection goes). This could have been more useful if the temperature range has been chosen wisely. For e.g., Brazil is one of the top maize-growing countries and the summer temperature goes beyond 40 °C. Even in some parts of China, the summer temperature goes beyond 30 °C. Now, with the available data, it is more of academic interest than practical use.
The abstract should give a well-constructed outcome of the study for the end users that this strain at this particular condition is highly efficient in toxin production, and should be managed well to avoid the production of this toxin.
Section 2.1: L103 shows aw of 0.995 is optimal; in the entire paper either it is 0.99 or 0.95 aw, so what 0.995 is indicating? Use uniform values throughout the manuscript.
Fig. 1: What are A1, A2, P1…..V2? Mention in figure caption.
Fig. 2: None of the graphs indicate multiple comparisons among the treatments.
Fig. 7: Why this experiment was conducted only with F. verticilloides, when other 2 strains also produced FB1?
Fig. 8: A lot of abbreviations have been used, please explain these abbreviations in the Fig caption. Readers will not search through the entire 25-page manuscript to find the details of abbreviations used in any figure.
Fig. 9: Perform non-parametric analysis for analyzing gene expression, as it is not parametric data.
Material and Methods: It must be explained in detail to ensure reproducibility.
Specific comments for M&M: Table 1: Details of cultures accessioned in any culture collection should be given along with their accession in genebank (EMBL/DDBJ/NCBI)
Section 4.2: L416- Mention how much % of glycerol needs to be added to reach these aw levels in PDA
L420: What is the basis for choosing these temp. Ranges
L421: Correct to-“ 7th, 14th, and 21th days.”
Material and methods should be clearly explained. L424 says -the entire mycelia were removed on the 7th day, then how are the 14th and 21st-day data taken?
L421 says that 3 replications were used in this experiment and if these were utilized in the L425 experiment, how you got the experimental material for the L428 experiment? You have already used those three plates for biomass purposes.
Again, L449 is destructive sampling which is mutually exclusive of the L425 experiment. If you have isolated mycelia for RNA extraction from 3 biological replicates, how you used the same for biomass calculation? And nowhere it has been mentioned that three different sets of experiments were designed separately for biomass, toxin extraction for LC-MS, and for RNA extraction expt. This need at least 9 replication of the initial study, of which 3 each would be used for biomass, toxin extraction for LC-MS, and for RNA extraction expt. Either it is written wrong, or conducted wrong; authors need to very critically reply to this query before publication.
L472: Whether you have used ‘2ˆ(–delta delta Ct)’ or only ‘delta delta Ct’? clarify

Author Response
Response to Reviewer 2 Comments
Point 1: Section 2.1: L103 shows aw of 0.995 is optimal; in the entire paper either it is 0.99 or 0.95 aw, so what 0.995 is indicating? Use uniform values throughout the manuscript.
Response 1: It is our mistake. 0.99 is right. We have made the corrections.
Point 2: Fig. 1: What are A1, A2, P1…..V2? Mention in figure caption.
Response 2: A1, A2, P1…..V2 means the corresponding strain. We gave the explanation in the table 1 and mentioned these in figure caption according to your suggestion.
Point 3: Fig. 2: None of the graphs indicate multiple comparisons among the treatments.
Point 3: We did perform multiple comparisons among the treatments in every figure based on two points: First, our main aim is to determine the effect of environment conditions on toxin levels and We conduct ANOVA to check the influence. We focus on the largest values of each strain strain and sometimes there are not significant difference among the treatments. Second, there are more parameters in part graphs, if we add various letters to show the significance of difference, the figure is confusing.
Point 4: Fig. 7: Why this experiment was conducted only with F. verticilloides, when other 2 strains also produced FB1?
Response 4: It is our mistake leading to the confused figure caption . Two strains producing FB1 belong to F. verticillioides. We have made the corrections in all figures.
Point 5: Fig. 8: A lot of abbreviations have been used, please explain these abbreviations in the Fig caption. Readers will not search through the entire 25-page manuscript to find the details of abbreviations used in any figure.
Response 5: We have explained these abbreviations in the figure caption according to your suggestion.
Point 6: Fig. 9: Perform non-parametric analysis for analyzing gene expression, as it is not parametric data.
Response 6: Non-parametric analysis for analyzing gene expression was performed according to the suggestion. The results were shown in Table 6 and 7.
Point 7: Specific comments for M&M: Table 1: Details of cultures accessioned in any culture collection should be given along with their accession in genebank (EMBL/DDBJ/NCBI)
Response 7: Our strains were not given the accession in genebank, maybe we can perform this work in the future. Our strains were isolated from the local maize samples and can reflect the physiological characteristics.
Point 8: Section 4.2: L416- Mention how much % of glycerol needs to be added to reach these aw levels in PDA
Response 7: The amount of glycerol was added in the manuscript.
Point 9: L420: What is the basis for choosing these temp. Ranges
Response 9: In our previous study, we found 25 and 30 °C were the optimal temperature for culture and the common condition for research. The growth at 35 °C was slower that at 20 °C. In addition, some strains were isolated from the samples in northern China where the average annual temperature was below 20 °C. We think the range of temperature was close to the field situation.
Point 10: L421: Correct to-“ 7th, 14th, and 21th days.”
Response 10: We have made the correction according to the suggestion.
Point 11: L424 says -the entire mycelia were removed on the 7th day, then how are the 14th and 21st-day data taken?
Response 11: There were no data came from the mycelia on the 14th or 21st-day, as the biomass and RNA were all generated from mycelia on the 7th day.
Point 12: L421 says that 3 replications were used in this experiment and if these were utilized in the L425 experiment, how you got the experimental material for the L428 experiment? You have already used those three plates for biomass purposes.
Response 12: When we performed these experiment, five replicates were applied in order to avoid the contamination of fungi or bacteria. However, I think three replication is possible. Mycelium from the same plate was firstly weighed as the biomass and then used for RNA extraction. The plate without mycelium was used in toxin analysis. The samples for biomass, toxin and RNA came from the same plate so that we can ensure data consistency and correspondence.
Point 13: Again, L449 is destructive sampling which is mutually exclusive of the L425 experiment. If you have isolated mycelia for RNA extraction from 3 biological replicates, how you used the same for biomass calculation? And nowhere it has been mentioned that three different sets of experiments were designed separately for biomass, toxin extraction for LC-MS, and for RNA extraction expt. This need at least 9 replication of the initial study, of which 3 each would be used for biomass, toxin extraction for LC-MS, and for RNA extraction expt. Either it is written wrong, or conducted wrong; authors need to very critically reply to this query before publication.
Response 13: The same response as above. When we performed these experiment, five replicates were applied in order to avoid the contamination of fungi or bacteria. However, I think three replication is possible. Mycelium from the same plate was firstly weighed as the biomass and then used for RNA extraction. The plate without mycelium was used in toxin analysis. The samples for biomass, toxin and RNA came from the same plate so that we can ensure data consistency and correspondence.
Point 14: L472: Whether you have used ‘2ˆ(–delta delta Ct)’ or only ‘delta delta Ct’? clarify
Response 14: 2-△△Ct method was applied. We have made the correction.
Reviewer 3 Report
It is a very interesting paper, with much work done, however I think tehre are many mistakes in writing... english sounds informal in many sentences
The abstract needs a carefull revision.
line 7: re- write
line 10-11: it could be imporved
line 12: ...while the former species???
key contributions: subindice of W in aW, ..... genes in strains within the FFSC...
introduction: in the first paraghraph there are no cites...
maize is always under attack.. maybe maize is susceptible to...
line 35: predominant... due to its effect on production? or important due to..?
line 50: food? or maize?
line 51: delete "the"
materials and methods
Table 1 in this journal it the last table, also in the tittle, Fusarium fujikuroi should be in italics
mycotoxin analysis: you dot write LOQ, LOD values.
line 100. after Figure 1 a dot. a then. All strains...
line 104: both numbers? both variables?
line 105: ...were reduced, also all strains were able to...
line 107: biomass "acumulation"?
line 106-108: it should be re written
table numbers should be changed
figure 1 leyend: It should be better to include temp and aW data in the figure, teh figure leyend must be self-explanatory. after each species name you shoul put (A1, A2....)
table 2, 3, 4 and check others, the tittle: ....water activity, delet "and", insert a comma, temperature (T), and their interactions...
Avoid abbreviations in tables tittles, Fusarium fujikuroi species complex.
line 124: whithin FFSC evaluated strains, delete "while", it was...
line 127: than than (twice), re write the sentence.
line 129: both F. fujikuroi? re write.
linq 192: on day 21? or.. at 21 incubatoin days?
line 193: the 2 evaluated strains.
line 197: delete "between different variables".
table 6: tittle: .... and their interactions....., on .... by Fusarium andiyazi (in full)
lines 202: are strong?, were?
figure 8 leyend: should be re written
table 7 is ok?, in the text you mention an ANOVA of two factors (aw and T), tehn in the table the factor S is included.. so?, aldo strains names are not ok.
lines 237-238- Re write. the genes were selected to evaluates teh effect of abiotic factors on...?
line 245:?
line 250: after the dot. 20 should be written in full: Twenty
line 255: great?
line 291-294: it should be re written, very long sentence.
Author Response
Response to Reviewer 3 Comments
Point 1: The abstract needs a careful revision.
Response 1: According to the suggestion, we have revised this part.
Point 2: line 7: re- write
Response 2: This sentence was rewritten.
Point 3: line 10-11: it could be improved.
Response 3: This sentence was improved.
Point 4:line 12: ...while the former species???
Response 4: the former species is F. andiyazi. We have rewritten this sentence.
Point 5: key contributions: subindice of W in aW, ..... genes in strains within the FFSC...
Response 5: We have made the correction according to the suggestion.
Point 6: introduction: in the first paragraph there are no cites...
Response 6: We have added the reference in the first paragraph.
Point 7: maize is always under attack.. maybe maize is susceptible to...
Response 7: We have revised this sentence according to the suggestion.
Point 8:line 35: predominant... due to its effect on production? or important due to..?
Response 8: We have revised predominant to important.
Point 9:line 50: food? or maize?
Response 9: We have revised food to maize.
Point 10: line 51: delete "the"
Response 10: We have revised this part according to the suggestion.
Point 11: Table 1 in this journal it the last table, also in the tittle, Fusarium fujikuroi should be in italics
Response 11: The sequence of tables was updated. We have revised the title of the table.
Point 12: mycotoxin analysis: you dot write LOQ, LOD values.
Response 12: We have added LOQ and LOD values in materials and methods.
Point 13: line 100. after Figure 1 a dot. a then. All strains...
Response 13: We have revised the sentence according to the suggestion.
Point 14: line 104: both numbers? both variables?
Response 14: We have revised both numbers to both variables.
Point 15: line 105: ...were reduced, also all strains were able to...
Response 15: We have revised the sentence according to the suggestion.
Point 16: line 107: biomass "accumulation"?
Response 16: We have made the correction according to the suggestion.
Point 17: line 106-108: it should be re written
Response 17: The sentence was rewritten.
Point 18: table numbers should be changed
Response 18: The sequence of tables was updated.
Point 19: figure 1 legend: It should be better to include temp and aW data in the figure, the figure
legend must be self-explanatory. after each species name you should put (A1, A2....)
Response 19: We have added the corresponding code of the each strain after each species name.
Point 20: table 2, 3, 4 and check others, the tittle: ....water activity, delete "and", insert a comma, temperature (T), and their interactions...
Response 20: We have made the correction according to the suggestion.
Point 21: Avoid abbreviations in tables tittles, Fusarium fujikuroi species complex.
Response 21: We have revised the table tittles according to the suggestion.
Point 22: line 124: within FFSC evaluated strains, delete "while", it was...
Response 22: We have revised this sentence according to the suggestion.
Point 23: line 127: than than (twice), re write the sentence.
Response 23: We have made the correction.
Point 24: line 129: both F. fujikuroi? re write.
Response 24: We have revised this sentence.
Point 25: line 192: on day 21? or.. at 21 incubatoin days?
Response 25: We have revised this part according to the suggestion.
Point 26: line 193: the 2 evaluated strains.
Response 26: We have revised this part according to the suggestion.
Point 27: line 197: delete "between different variables".
Response 27: We have deleted these words.
Point 28: table 6: tittle: .... and their interactions....., on .... by Fusarium andiyazi (in full)
Response 28: We have revised the table tittles according to the suggestion.
Point 29: lines 202: are strong?, were?
Response 29: We have revised are to were.
Point 30: figure 8 legend: should be re written
Response 30: Legend of figure 8 were rewritten.
Point 31:table 7 is ok?, in the text you mention an ANOVA of two factors (aw and T), then in the table the factor S is included.. so?, also strains names are not ok.
Response 31: We have made the correction.
Point 32: lines 237-238- Re write. the genes were selected to evaluates the effect of abiotic factors on...?
Response 32: We have revised this sentence according to the suggestion.
Point 33:line 245:?
Response 33: We have deleted this sentence.
Point 34: line 250: after the dot. 20 should be written in full: Twenty
Response 34: We have revised this sentence.
Point 35: line 255: great?
Response 35: We have deleted this sentence.
Point 36:line 291-294: it should be re written, very long sentence.
Response 36: We have revised this part to two sentences.
Reviewer 4 Report
The “Effect of abiotic conditions on growth, mycotoxin production and gene expression by Fusatium fujikuroi species complex strains from maize” article is of much interest to determine the effects of water ability, temperature, and strain in the mycotoxin production. The article has a well explained introduction, material and method, results, and discussion.
Material and methods are well explained and are enough for the aim of this study. Although, the morphological identification of fungal strains is ok and molecular identification of the strain is considerable enough information to determine species within the Fusarium fujikuroi species complex as referenced in the manuscript.
I have read and try to stay focus on the develop of the manuscript and have found some references mistakes that may need a doble check. (Different formats).
Figure on the results with different interaction of strain and abiotic conditions are a little bit small, but the main information is well explained at the statistics tables under it.
I recommend update of the cites and references in order to have new information. And may be, as a suggestion I will be interested in adding some mycotoxin regulation of china, sud America, Europe, and more.
Best of lucks!
Author Response
Response to Reviewer 4 Comments
Point 1: Material and methods are well explained and are enough for the aim of this study. Although, the morphological identification of fungal strains is ok and molecular identification of the strain is considerable enough information to determine species within the Fusarium fujikuroi species complex as referenced in the manuscript.
Response 1: As an attachment, we have submitted EF1 alpha sequences of the strains used in this study for molecular identification.
Point 2: I have read and try to stay focus on the develop of the manuscript and have found some references mistakes that may need a double check. (Different formats).
Response 2: Thanks for the advice. We have checked the references again and made the corrections.
Point 3: Figure on the results with different interaction of strain and abiotic conditions are a little bit small, but the main information is well explained at the statistics tables under it.
Response 3: We will follow the suggestion of the editor about the figure on the results.
Point 4: I recommend update of the cites and references in order to have new information. And may be, as a suggestion I will be interested in adding some mycotoxin regulation of china, sud America, Europe, and more.
Response 4: We have added several reviews about the mycotoxin regulation worldwide according to the suggestion. Due to the length and research content, we did not list the limit standards and just briefly described the situation regulations relating to mycotoxins in grains.
Round 2
Reviewer 3 Report
I think the manuscript have been improved, however a final lecture of it should be done in order to detect some minnor detains. For example, the second sentence of the introduction still need a cite. The first sentence of the second paraghraph in the introduction... the most important??? So i think that authors should read carefully the manuscript and revised every sentence.
Author Response
Response to Reviewer 3 Comments
Point 1: the second sentence of the introduction still need a cite.
Response 1: The references were added in the first paragraph.
Point 2: The first sentence of the second paraghraph in the introduction.
Response 2: We have revised this sentence.
Point 3: a final lecture of it should be done in order to detect some minnor detains.
Response 3:We have checked every sentence and made several corrections.